# Distinct roles of amylin and oxytocin signaling in intrafamilial social behaviors at the medial preoptic area of common marmosets

Takuma Kurachi[1,2,9], Kazutaka Shinozuka[1,9], Chihiro Yoshihara[1,3], Saori Yano-Nashimoto [1,4], Ayako Y. Murayama[1,5], Junichi Hata[5,6,7], Yawara Haga[6], Hideyuki Okano [5,6] & Kumi O. Kuroda [1,3,8 ✉]

Calcitonin receptor (Calcr) and its brain ligand amylin in the medial preoptic area (MPOA) are found to be critically involved in infant care and social contact behaviors in mice. In primates, however, the evidence is limited to an excitotoxic lesion study of the Calcr-expressing MPOA subregion (cMPOA) in a family-living primate species, the common marmoset. The present study utilized pharmacological manipulations of the cMPOA and shows that reversible inactivation of the cMPOA abolishes infant-care behaviors in sibling marmosets without affecting other social or non-social behaviors. Amylin-expressing neurons in the marmoset MPOA are distributed in the vicinity of oxytocin neurons in the anterior paraventricular nucleus of the hypothalamus. While amylin infusion facilitates infant carrying selectively, an oxytocin's inverse agonist, atosiban, reduces physical contact with non-infant family members without grossly affecting infant care. These data suggest that the amylin and oxytocin signaling mediate intrafamilial social interactions in a complementary manner in marmosets.

[1] Laboratory for Affiliative Social Behavior, RIKEN Center for Brain Science, Saitama, Japan. [2] Department of Agriculture, Tokyo University of Agriculture and Technology, Tokyo, Japan. [3] School of Life Science and Technology, Tokyo Institute of Technology, Kanagawa, Japan. [4] Laboratory of Physiology, Department of Basic Veterinary Sciences, Graduate School of Veterinary Medicine, Hokkaido University, Hokkaido, Japan. [5] Department of Physiology, Keio University School of Medicine, Tokyo, Japan. [6] Laboratory for Marmoset Neural Architecture, RIKEN Center for Brain Science, Saitama, Japan. [7] Graduate School of Human Health Sciences, Tokyo Metropolitan University, Arakawa, Tokyo, Japan. [8] Laboratory for Circuit and Behavioral Physiology, RIKEN Center for Brain Science, Saitama, Japan. [9] These authors contributed equally: Takuma Kurachi, Kazutaka Shinozuka. ✉email: kurodalab@bio.titech.ac.jp

Maternal care is essential for the growth of mammalian infants. As such, mothers possess innate brain circuitry that enables them to nurture the young[1–3]. While mothers are typically the sole caregivers of infants in most mammals, in several species, including laboratory mice, common marmosets, and humans, infant care is shared among family members such as fathers and older siblings[4]. The medial preoptic area (MPOA), positioned anterior to the hypothalamus in the basal forebrain, is the hub of maternal, paternal, and alloparental caregiving behaviors. Its role was first identified in rats and has since been supported by studies in several mammalian species (see refs. [3,5] for a comprehensive review of this literature). MPOA neurons expressing estrogen receptor α or galanin have been shown to mediate pup retrieval and pup grooming, respectively[6–10].

Our laboratory narrowed down the key MPOA subregion into the central part of the MPOA (cMPOA)[11], which overlaps with the lateral subdivision of the medial preoptic nucleus. Within the cMPOA, calcitonin receptor (Calcr) expressing neurons play a critical role in maternal and alloparental behaviors in mice[12,13]. Calcr is a Gq-coupled transmembrane receptor activated by binding to calcitonin in the periphery. However, in the brain, Calcr forms a complex with Receptor Activity Modifying Proteins (Ramps) and binds to amylin instead of calcitonin[14]. Amylin/IAPP (islet amyloid polypeptide) is a brain–gut peptide produced in the pancreas, hindbrain, hypothalamus and the MPOA[14,15]. MPOA Amylin expression is increased in the postpartum period and functions in maternal adaptation via its action on Calcr in rodents[13,16–18]. Furthermore, MPOA amylin expression is maintained by group living and facilitates contact-seeking behaviors to maintain group living in female mice[18]. Female mammals are typically more sociable than males (except family-living primates) plausibly for the benefits of maternal care[19,20]. Thus, amylin-Calcr signaling in the MPOA appears critical for parental care and sociability.

We investigated the brain mechanism for primate parenting using the New World monkey common marmoset, *Callithrix jacchus*. Common marmosets live in a family of a breeding pair (mother and father) and their offspring, occasionally with unrelated adults in the wild[21]. Infants are generally born as twins, and infant care is a significant burden for this arboreal species[22,23]. The family uses elaborate vocal communication to share the infant carrying among members. These features make the common marmosets a valuable model of family parenting in humans[24,25]. We have found that the marmoset counterpart of mouse cMPOA contains Calcr-expressing neurons that are transcriptionally activated during infant care[23]. The excitotoxic lesions at the marmoset cMPOA, which eliminate neuronal cell bodies sparing local passing fibers, turned alloparental caregivers (siblings) rejective to infant carrying and dampened the total infant carrying without affecting general health, locomotion, feeding, and other daily activities. The similar-sized lesions in the posterior septum, just dorsal to the marmoset cMPOA, did not significantly disturb infant carrying while affecting general locomotion[23]. These data suggest that the marmoset cMPOA has evolutionarily conserved neuronal properties and functions in infant care. However, while these permanent lesion-behavior mappings have provided valuable insights, they also have inherent limitations; for example, permanent brain dysfunctions may induce functional adaptation and remodeling of the neighboring brain areas[26,27]. They thus should be complemented with data from reversible suppression of the function.

In the present study, we aimed to reversibly manipulate the marmoset cMPOA neurons and their molecular signaling and assess behavioral outcomes in infant-directed and other intrafamilial behaviors. We used pharmacological manipulation via

chronically implanted cannula at the cMPOA, as the viral vector-mediated genetic strategies are still under development in the marmoset MPOA (see ref. [23]).

## Results

### Study design of pharmacological manipulation of the cMPOA.
To investigate the role of Calcr-amylin signaling in the MPOA in various social behaviors in marmosets, we used sibling siblings of the infants for invasive experiments as in our previous study[23] (Table 1). The reasons for using siblings instead of breeding parents were threefold: first, using parents for invasive studies may affect family sustenance and infant survival; second, older siblings, especially those who experienced infant care at least once previously, provide comparable infant carrying as parents, regardless of sex[23]; and finally, the responsible neural circuit for maternal, paternal and alloparental behaviors are shown to be essentially the same in the rodent studies[3,13,28,29].

To control the possible side effects of bilateral guide cannula implantation (see Methods), we employed two types of surgical experiments; the first batch of subjects (Fig. 1a, top and Table 1) received a unilateral infusion of n-methyl-d-aspartic acid (NMDA), which resulted in excitotoxic neuronal ablation sparing passing fibers[30,31], and the contralateral implantation of guide cannulas. The second batch of sibling subjects received bilateral implantations of guide cannulas targeting the cMPOA (Fig. 1a, bottom). In general, unilateral inputs alone can drive basic instinctive behaviors; we initially presumed that inactivation of the cMPOA in these two groups should result in a similar result, provided the side effects of the surgery were minimal. These surgeries were performed during the late gestational period of the mother, and the subject siblings were returned to the family cage after the 2-day recovery period (Fig. 1b). All subjects could return to the family after the surgery, except for one that received attacks from the original family and was excluded from the study.

After the parturition of the mother (between PND 1 and 4), the subject siblings were first confirmed for their intact infant-care behaviors without infusion through the cannula. Then they were tested for infant-directed and other behaviors (Table 2) in the dyadic infant-retrieval assay (Fig. 1c) and within the family (family observation) (Fig. 1d) under the influence of pharmacological treatment at the cMPOA (Table 1). These behavioral experiments were performed using two adjacent compartments of their family cage complex, consisting of three cages connected via holes and a tunnel in front of the cage.

### Transient inactivation of the MPOA by muscimol abolishes infant care.
We first examined the effects of transient cMPOA inactivation by muscimol (GABA-A receptor agonist) on various marmoset behaviors (Figs. 2a–c and 3a–p and Supplementary Tables 1 and 2). The initial phase of the experiments suggested that when the muscimol concentration was high, the muscimol infusion into the cMPOA induced drowsiness, which might involve the wake-promoting neurons in the ventral MPOA[32]. Thus, we titrated the muscimol concentration and infusion volume and determined the maximal concentration that did not affect wakefulness, locomotion and other general activities for each animal (designated as C100), which are summarized in Table 1 (for example, 0.4–0.6 μg/μl of 500 nl muscimol solution for unilateral infusion with the contralateral lesion). By alternating the muscimol and vehicle injections in the same animals, the effects of muscimol were shown to be reversible and completely washed out by the next day (Fig. 2d).

We found that C100 muscimol infusion into the cMPOA significantly increased retrieval latency and rejection rate and decreased total infant-carrying rate in the dyadic infant-retrieval

**Table 1 The list of subjected marmosets.**

| subject | sex | family | age at first assay (mo) | previous infant-care | old sib | young sib | infant | cannula | coordinate | muscimol C100 (µg/µl) | muscimol dose (nl) | AC187 conc. (µg/µl) | atosiban conc. (ng/µl) | muscimol (interval / order) | amylin (interval / order) | AC187 (interval / order) | atosiban (interval / order) |
|---|---|---|---|---|---|---|---|---|---|---|---|---|---|---|---|---|---|
| Kotaro | M | A | 23 | y | 1 | 2 | 2 | unilat. | atlas | 0.4 | 500 | | | 2 h FR | | | |
| Myoga | M | B | 17 | y | 1 | 2 | 2 | unilat. | atlas | 0.6 | 500 | | | 2 h FR | | | |
| Namihei | M | C | 12 | y | 0 | 2 | 2 | unilat. | atlas | 0.6 | 500 | | | 2 h FR | | | |
| Dahlia | F | D | 22 | y | 0 | 2 | 2 | unilat. | MRI | 0.6 | 500 | | | 2 h FR | | | |
| Hanako | F | A | 23 | y | 1 | 2 | 2 | unilat. | atlas | 0.4 | 500 | | | 2 h FR | | | |
| Nabe | F | E | 30 | y | 0 | 2 | 2 | unilat. | MRI | 0.5 | 500 | 10/20 | | 1 h FR | 2 h FR | 1 h FR/RF | 30 m F |
| Ringo | F | F | 28 | y | 1 | 2 | 2 | unilat. | MRI | | | | | | 30 m RF | 2 h FR | |
| Senae | F | G | 26 | y | 1 | 1 | 2 | unilat. | MRI of littermate | | | 10 | | | | 2 h FR | |
| Brian | M | H | 12 | no | 1 | 0 | 1 | bilat. | atlas | 0.4 | 200 | 20 | 50 | 30 m RF | | 30 m RF | 30 m RF |
| Enoki | M | D | 37 | y | 1 | 0 | 1 | bilat. | MRI | 0.3 | 500 | | | 30 m RF | | | |
| Ken-ichi | M | G | 6 | no | 2 | 1 | 2 | bilat. | atlas | 0.4 | 200 | | 50 | | | | 30 m RF |
| Umihei | M | C | 14 | y | 1 | 1 | 2 | bilat. | MRI | 0.3 | 500 | 10/20 | 12.5/50 | 30 m RF | | 30 m FR/RF | 30 m RF |
| Cookie | F | E | 19 | y | 1 | 2 | 2 | bilat. | MRI | 0.4 | 100 | 20 | | 30 m RF | 30 m RF | 30 m RF | |
| Fuyumi | F | G | 12 | y | 2 | 1 | 2 | bilat. | MRI | 0.4 | 100/200 | 20 | | 30 m RF | 30 m RF | 30 m RF | |
| Hibari | F | G | 6 | no | 1 | 2 | 2 | bilat. | atlas | 0.4 | 200 | | 50 | 30 m RF | 30 m RF | | 30 m RF |
| Lemon | F | F | 26 | y | 1 | 2 | 2 | bilat. | MRI | 0.4 | 100 | 10/20 | | 30 m RF | 30 m RF | 30 m RF | |
| Rika | F | C | 14 | y | 1 | 1 | 2 | bilat. | MRI | 0.4 | 100 | 10/20 | | 30 m RF | 30 m RF | 30 m FR/RF | |

FR: Family observation → Retrieving assay
RF: Retrieving assay → Family observation

FR: Family observation
RF: Retrieving assay

assays in both unilateral and bilateral experiments (Fig. 2a–c). The generalized linear mixed model (GLMM) analyses revealed significant main effects of muscimol versus the vehicle infusion and no main effect of the surgery type. The interactions between the muscimol and cannula types are found in the retrieval latency and carrying rate, possibly due to the partial defects in infant-directed behaviors observed in the animals that received unilateral vehicle infusion with the contralateral cMPOA lesion.

In family observations, we did not observe significant effects of muscimol microinfusions in infant-directed behaviors (Fig. 3a, b and Supplementary Table 2). This is at least partly due to the outlier session of Rika, who was huddled with other family members during most of the C100 session (Fig. 3d, high "body contact" rate in C100). The infant was moved across the huddled members, including Rika, thus causing high "retrieval" and "carrying" rates without Rika's voluntary actions (Fig. 3a, b). When excluding Rika's sessions, the carrying rate was significantly decreased by muscimol microinfusion into the cMPOA (Supplementary Table 3).

Of additional note, we examined the sex difference of the effects of muscimol using the models including sex as an explanatory parameter together with drug and cannula type and their interactions. We did not observe apparent sex differences and interactions in most behaviors except those with very low occurrence frequency (Supplementary Tables 4 and 5), in harmony with our previous study[23]. Therefore, we adopted the models excluding the sex effect.

**The selectivity of cMPOA functions in infant-care behaviors among other social and non-social behaviors of sibling marmosets within the family.** In contrast to the drastic effects of cMPOA muscimol infusions in the infant-retrieval assays, the cMPOA muscimol infusion did not significantly affect intrafamilial behaviors except self-grooming (Fig. 3o and Supplementary Tables 2 and 3). C100 muscimol did not alter social contact with family members other than infants (Fig. 3d), indicating that the decrease in infant carrying was not caused by a general avoidance of social interactions. No agonistic interactions with other family members were observed. Thus, the increased infant rejection was not driven by increased general aggression.

**Histological analysis of the marmoset MPOA.** Before testing the involvement of amylin-Calcr signaling in the infant-care behaviors in marmosets, we examined the molecular expression of amylin in the MPOA and the amylin's relative localization with the parenting-induced c-Fos expression (Fig. 4).

Calcr-immunoreactive (-ir) neurons are distributed in a discrete centromedial subregion of the posterior (Interaural +9.8) marmoset MPOA, where c-Fos-ir neurons are also distributed after infant care (Fig. 4a). The rate of colocalization of Calcr and c-Fos was ~30% of all Calcr+ neurons in marmosets, while 60% in mice, which might be due to the relative difficulty of marmoset brain histochemistry than in mice.

In the coronal section at Interaural +9.0 (Fig. 4b), the cluster of c-Fos-ir neurons is located more medially in the vicinity of the oxytocin-expressing neurons (as evidenced by the expression of NeurophysinI (NPI), the oxytocin carrier protein that is encoded in the same gene as oxytocin) in the anterior subdivision of the paraventricular nucleus of the hypothalamus (aPVH)[33]. Parenting-induced c-Fos-ir and NPI-ir do not colocalize in the alloparenting marmoset brain, as in mice[11,34].

Consistent with the mouse findings, Amylin-expressing neurons were distributed at the posterior cMPOA and the aPVH (Fig. 4c, d). Like other bioactive neuropeptides, subcellular localization of amylin depicted by an anti-amylin antibody

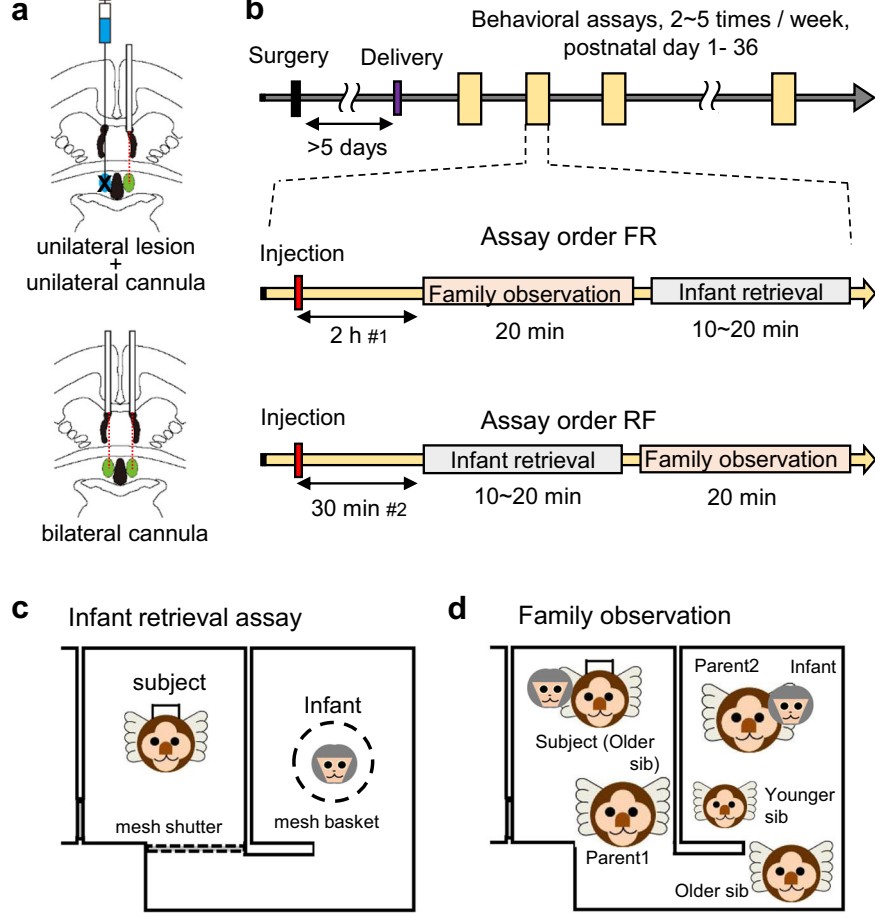

**Fig. 1 Experimental designs. a** A schematic illustration of two types of surgery. Top: the unilateral lesion by NMDA injection into the right cMPOA, and contralateral guide cannula implantation below the corpus callosum in the left hemisphere. Bottom: a dual-cannula was implanted at the level of the ventral surface of the corpus callosum. Red dotted lines: the internal cannula path. In this illustration, the figures of coronal brain section were traced from marmoset brain atlas[33,60]. **b** Timelines of the two types of experiments. We used the top timeline for the initial experiments to observe intact family behaviors without perturbation associated with dyadic infant-retrieval assay. We used the bottom scheme for amylin infusion to observe the infant-retrieval assay with the highest amylin concentration, considering the high degradation rate of amylin peptide in the brain tissue[61]. **c** A schematic illustration of infant-retrieving assay. After the shutter opens, the subject can freely interact with the infant in the basket. Although the basket is open, infants younger than 1 month cannot freely move by themselves, so the subject approach, contact the infant, and allow the infant to cling. The latency to the infant clinging (retrieval latency) was recorded, and the behaviors of the subject were observed further for 10 min with 30-s bins. **d** A schematic illustration of family observation assay. Various behaviors of family members (listed in Table 2) were coded for 20 min with 30-s bins.

(Fig. 4c) was found mainly in the cis Golgi complex[18], resulting in dotty signals in the neuronal soma. Amylin mRNA distribution detected by RNAscope is consistent with the amyin-ir (Fig. 4d). Overall, the parenting-relevant cMPOA – aPVH neuroanatomy in marmosets is similar to that in mice, although there are some differences in the spatial distributions of each neuronal type.

**Amylin reduces infant rejection and facilitates infant carrying**. To examine the effect of Amylin signaling on infant care and other behaviors, amylin or its control (vehicle, aCSF) was infused at the cMPOA in the second or third postnatal week, when the carrying rate of siblings started to decline (Fig. 5a–c). Compared to the control, amylin infusion significantly decreased the rejection rate and increased the carrying rate, without affecting the retrieval latency (Fig. 5a–c and Supplementary Table 6).

As an opposite manipulation, a Calcr-antagonist AC187 was infused at the cMPOA during the early postnatal period. The increase of the rejection rates after AC187 or vehicle infusions was close to significance (Fig. 5b, $p = 0.0596$, $t = 2.245$, df = 7). The retrieval latency and the carrying rate after the AC187

infusion were not significantly different from that after the vehicle infusion (Fig. 5a, c and Supplementary Table 7).

Neither amylin nor AC187 infusions at the cMPOA significantly affected social and non-social behaviors in the family observation assay (Fig. 6a–p and Supplementary Table 7). Notably, although amylin is a well-known mediator of satiety, pharmacological manipulations of amylin signaling in the cMPOA did not affect feeding behavior, further supporting the selectivity of the cMPOA amylin-Calcr signaling in infant care.

**An inverse-agonist of oxytocin reduces social contact of sibling marmosets with family members other than infants**. The oxytocin system has been implicated in affiliative social contacts in rodents[35–38], while its role in (allo)parental care has been variable among previous literature[34,39–43]. Because oxytocin neurons are located close to amylin+, Calcr+ and parenting-induced c-Fos+ neurons in marmosets (Fig. 4), we examined the effects of an oxytocin's inverse agonist atosiban in intrafamilial behaviors of sibling marmosets. It turned out that atosiban infusion into the

**Table 2 List of observed behavior in family observation and infant-retrieving assays.**

| Sampling | Item | Description |
|---|---|---|
| Scan sampling | Carrying | Infant(s) clung on body |
| | Body contact | Physical contact of body parts with other marmoset other than infant(s) |
| | Staying cage | Cage where animal is staying (right or center) |
| | Locomotion | Change the staying cage with the previous bin |
| 1-0 sampling | Successful retrieval | Retrieve infant(s) from other carrier or cage floor and start carry |
| | Failed retrieval | Same motion as successful retrieval but infant does not cling |
| | Anogenital licking | Lick infant's anogenital region to facilitate infant's excretion |
| | Rejection | Carrier scratches, bites infant(s), or rub infant(s) cage wall/floor |
| | Social play | Chase, chased by, or wrestle with other marmoset |
| | Object play | Manipulate objects by hand or bite objects |
| | Grooming other | Manipulate other marmoset's fur by hand or teeth |
| | Being groomed | Groomed from other marmoset |
| | Self-grooming | Groom own body parts |
| | Self-scratch | Scratch own body parts by hand |
| | Scent marking | Rub anogenital area on cage |
| | Eating | Masticate food pellets |
| | Drink | Lick a water nozzle |
| | Vocalizing | Emit vocalization including twitter, phee, trill, tsik, ek, chatter, or vhee |
| | Stereotypy | Repeated vertical circling more than three times |
| | Yawn | Yawning |

Scan sampling: if one behavior occurs at the start moment of a 30-s bin, this behavior is regarded to exist throughout the 30-s bin. (i.e., If the behavior occurs and ends within a single bin, this behavior is not regarded to occur within the bin).
1-0 sampling: if one behavior occurs at any moment within the 30-s bin, this behavior is regarded to exist throughout the 30-s bin.

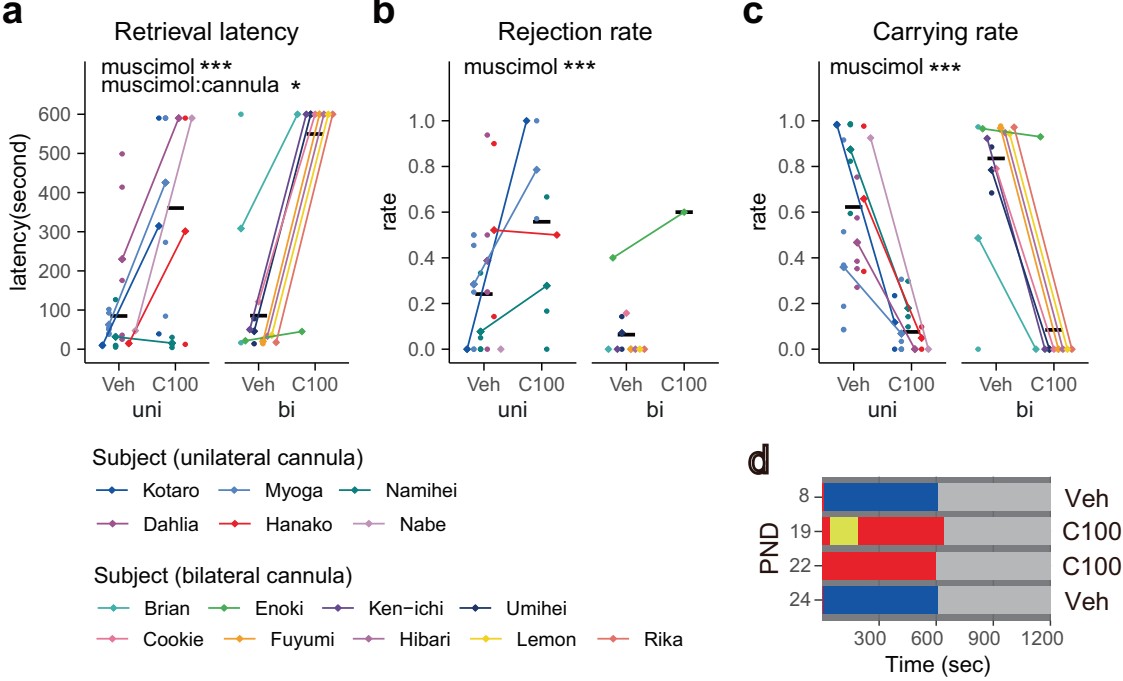

**Fig. 2 Transient cMPOA inhibition eliminates infant tolerance and sensitivity but no other social and non-social behaviors.** Retrieval latency (**a**), rejection rate (**b**) and carrying rate (**c**) in infant-retrieval assays after the vehicle (Veh) or muscimol microinfusions at the optimal level (C100). A total 54 assays (Kotaro and Hanako, and Ken-ichi and Hibari were observed at the same time because they were littermate) were conducted using eight families. Each dot corresponds to the value from a single experiment. The means of each subject (diamonds) in each condition are connected by a line. The black horizontal lines show the mean values of each condition of all subjects. Asterisks denote statistical significance (GLMM, Veh; $n = 33$ observations of 15 subjects, C100; $n = 27$ observations of 15 subjects, ***$p < 0.001$, *$p < 0.05$, †$p < 0.1$). Note that the rejection rate can be calculated only when the subject ever retrieved the infant during the session, and thus, the sample size was smaller for the C100 condition ($n = 8$ retrieved sessions of 5 subjects) than for the vehicle condition ($n = 32$ retrieved sessions of 15 subjects). **d** Representative raster plots of infant-retrieving assay by Kotaro. Red: a subject did not carry an infant, Blue: a subject carried an infant. Yellow: a subject showed rejecting behavior toward an infant carried.

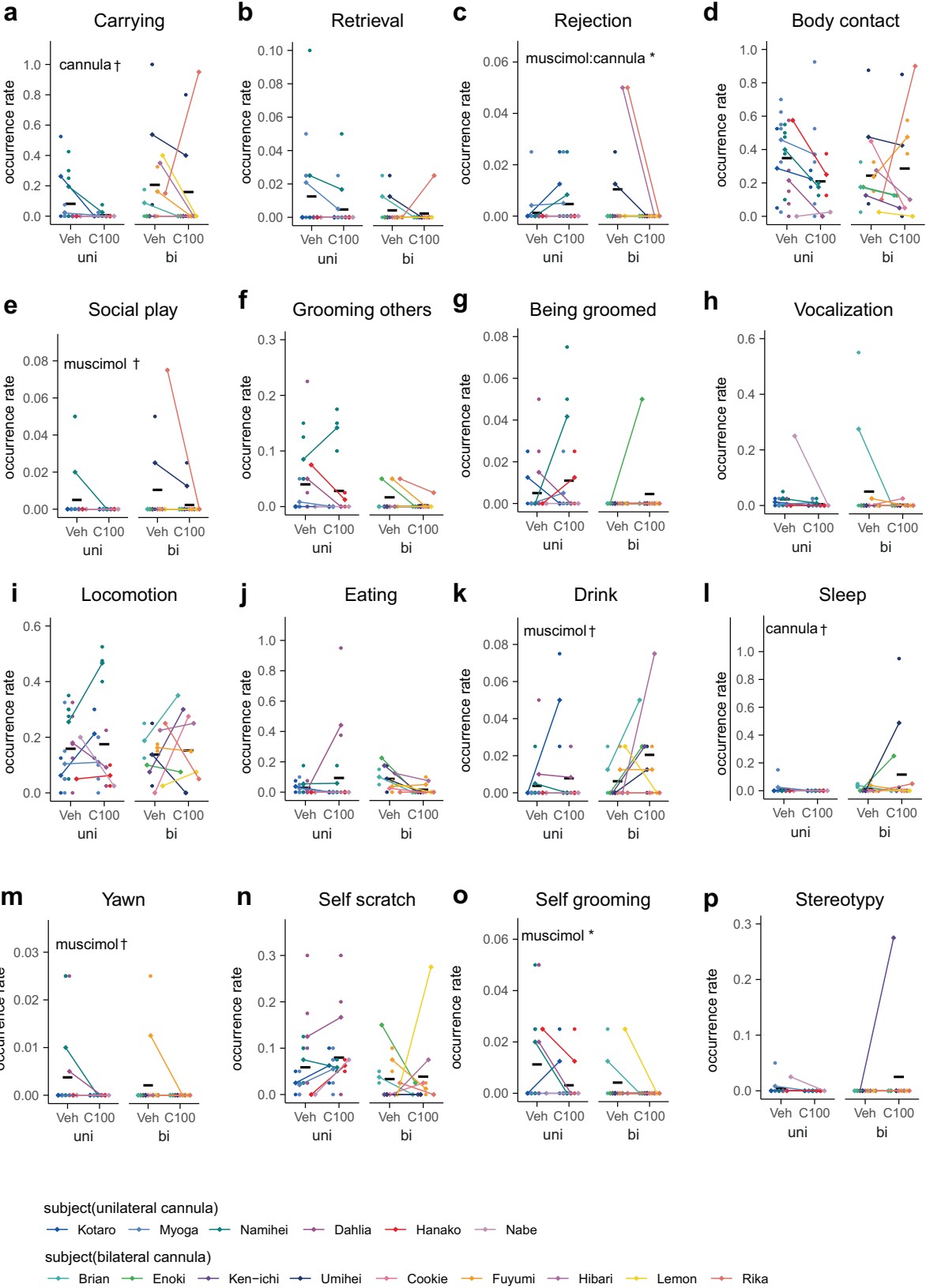

cMPOA robustly reduced body contact with non-infant family members (Fig. 7a–o and Supplementary Table 8). This amount of decrease in social contact was not explained by the increased eating behaviors, which is consistent with the known role of oxytocin in satiety[44]. Other social and non-social behaviors were

not affected by atosiban. Atosiban did not grossly disturb infant-directed behaviors, although the sample size was small due to the survival failure of Brian's younger siblings (Supplementary Fig. 2 and Supplementary Table 9). These data support results in rodents and suggest that the oxytocin signaling in the cMPOA is

**Fig. 3 Intrafamilial behaviors other than infant care are not grossly affected by the transient inhibition of the cMPOA.** Various behaviors observed in family observation (**a–p**) after the vehicle (Veh) or muscimol microinfusions at the optimal level (C100). A total of 54 assays (Kotaro and Hanako, and Ken-ichi and Hibari were observed simultaneously because they were littermates) were conducted using eight families. Each dot corresponds to the value from a single experiment. The means of each subject (diamonds) in each condition are connected by a line. The black horizontal lines show the mean values of each condition of all subjects. Asterisks denote statistical significance (GLMM, Veh; $n = 33$ observations of 15 subjects, C100; $n = 27$ observations of 15 subjects, ***$p < 0.001$, *$p < 0.05$, †$p < 0.1$). Among behaviors listed in Table 2, marking and object play are omitted because of low occurrence.

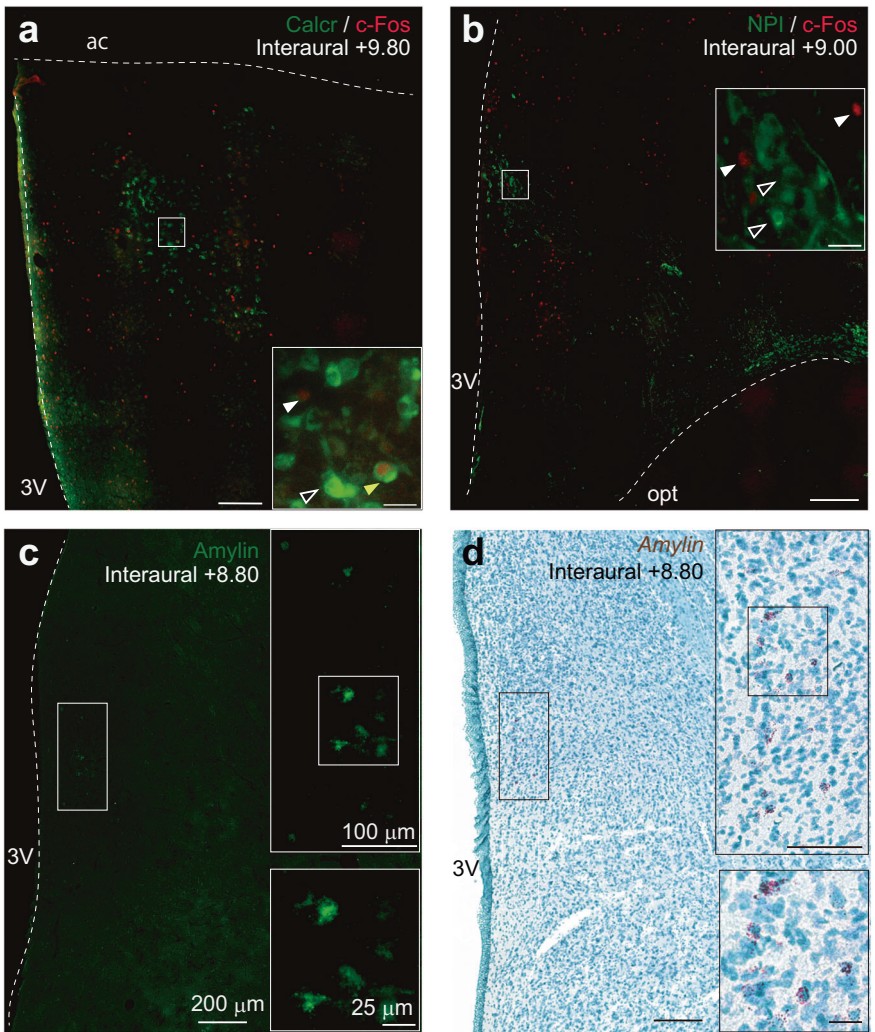

**Fig. 4 Chemical neuroanatomy of marmoset MPOA. a** Double immunohistochemical staining of c-Fos (red) and Calcr (green) in the cMPOA of alloparenting male sibling (Akira, postnatal day 425, infant exposure after 1 day of single housing, carried for 25/30 bins with 3-min rejection). Arrowheads: white, c-Fos-ir; black, Calcr-ir; yellow, double-ir neurons. **b** Double immunohistochemical staining of c-Fos (red) and NPI (green) in the aPVH of the same animal as in (**a**). Arrowheads: white, c-Fos-ir; black, NPI-ir. **c** Immunohistochemical staining of Amylin (green) in the aPVH of a group-housed adult female marmoset (Nana, postnatal day 762). **d** in situ hybridization staining using RNAscope of Amylin (black) in the aPVH of the same animal as in (**c**). An area framed in white and black was magnified. The scale bar sizes shown in (**c**) are the same for all the images.

more critical for social interactions with mature family members than infant care.

## Discussion

This study examined the effects of the temporary inactivation of cMPOA neurons on various behaviors and suggested the selective role of cMPOA in infant-care behaviors in sibling marmosets. Unlike neurotoxic cMPOA lesions[23], muscimol infusion increased the rejection rate and retrieval latency. It could be due to the possible recovery of the retrieval latency after permanent cMPOA lesions by compensatory mechanisms in other brain regions. Lecca et al.[45] recently demonstrated that the pathway from the bed nucleus of stria terminalis to the lateral habenula mediates aversion of pup distress vocalizations and thus facilitates pup retrieval in virgin female mice. Thus, retrieval sensitivity in response to infant distress vocalizations can be mediated not only by parental motivation via the cMPOA but also by aversion to infant calls via the lateral habenula or the bed nucleus of stria terminalis.

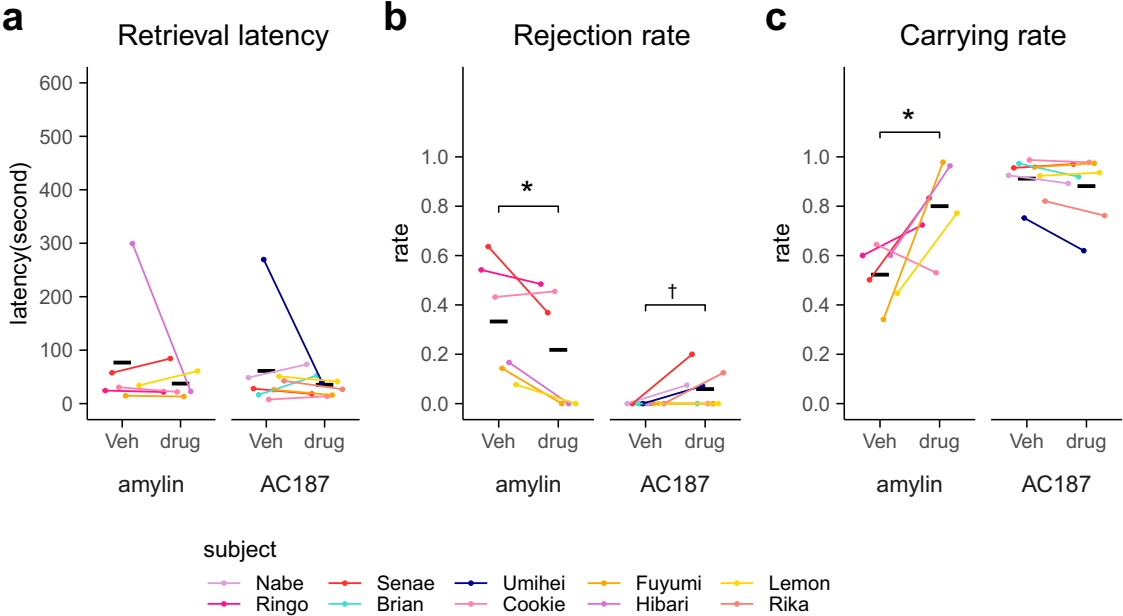

**Fig. 5 Effects of amylin and AC187 in the cMPOA on the infant-retrieving assay.** Retrieval latency (**a**), rejection rate (**b**) and carrying rate (**c**) in infant-retrieval assays after the vehicle (Veh), amylin or AC187 microinfusion. The infant-retrieving assay was conducted for 21 trials with amylin using 8 infants of 3 families and 24 trials with AC187 using 10 infants of 5 families. The means of each subject in each condition are shown as dots and connected between two conditions. The black horizontal lines show the mean values of each condition of all subjects. Asterisks denote statistical significance (paired $t$-test, amylin: control; $n = 11$ observations of 6 subjects, amylin; $n = 10$ observations of 6 subjects, AC187: control; $n = 12$ observations of 8 subjects, AC187; $n = 12$ observations of 8 subjects, *$p < 0.05$, †$p < 0.1$).

Anatomically, Calcr and amylin expression patterns in the marmoset MPOA are similar to those in mice[13,18], suggesting that the molecularly defined neural circuit in the MPOA is conserved across rodents and primates. However, the spatial organization of corresponding neuronal groups is not the same: in mice, the anterior extension of the aPVH is termed the anterior commissural nucleus (AC). The AC possesses distinctive features from the aPVH in cell morphology and its transcriptional activation in non-oxytocinergic neurons by performing parental care. Thus, the aPVH depicted in Fig. 4b in marmosets may functionally correspond to the AC in mice.

Functionally, amylin infusion into the cMPOA facilitates parental behaviors in marmosets. In mice, amylin infusion can activate Calcr+ neurons, and silencing of Calcr+ neurons disrupts parental care in virgin females and mothers[13]. These data suggest that amylin-Calcr signaling in the cMPOA promotes parental care, commonly in rodents and primates. The limitation of the present study is that the effects of AC187 infusion on infant-directed behaviors are inconclusive, leaving the possibility of the non-Calcr-mediated mechanisms for the amylin function, such as via calcitonin-receptor-like receptors. Still, we presented AC187 data as it is, because we believe that all experiments conducted at the cost of precious primate lives should be utilized. In the future, the cell-type or molecular-specific manipulation technique should become available in marmosets as in mice and may better elucidate the role of amylin-Calcr signaling in the MPOA.

In mice, contact-seeking among adult females also depends on amylin-Calcr signaling[18], supporting the notion that the primary driver of female group living in mammals is the benefits of maternal care[19]. However, in sibling marmosets, interactions with parents and non-infant siblings and communicative vocal behaviors are unaffected by permanent cMPOA lesions[23], temporary inactivation, or pharmacological cMPOA amylin-signal manipulations. Eibl-Eibesfeldt suggested that mammalian sociality has two principal motives: "parental drive" and "flight drive"[46]. We

have demonstrated that while parental motivation activates and depends on the cMPOA, defensive huddling against a sudden bright light (an example of "flight"-driven social contacts) does not in mice[18]. These findings collectively suggest that the intrafamilial social behaviors of sibling marmosets (targeted primarily toward their parents) may originate from a distinct motivation from infant care. The neural basis of such intrafamilial social motivation remains to be identified.

Oxytocin is a crucial regulator for milk ejection and uterine contraction throughout mammals[47] (see ref. [41]). This study has identified that oxytocin neurons are closely associated with the parenting-induced c-Fos expression in the cMPOA and the adjacent aPVH, similar to the findings for the AC in mice[11,34]. However, we could not detect a gross effect of suppressing oxytocin signaling in marmoset alloparental care, even though the same treatment significantly attenuated social contact with other family members (Fig. 7a–o and Supplementary Fig. 2). It may not be too surprising, as the previous studies yielded mixed results for the role of oxytocin in parental care of marmosets[48–50] and of rodents[34,39,41–43,51–54]. One possible explanation of this surprising discrepancy between female reproductive physiology triggered by oxytocin and mammalian parental care is that the evolutionary pressure for allomaternal care made oxytocin less important for infant-care motivation in species that cooperatively care for infants, such as laboratory mice, prairie voles and marmosets. In contrast, oxytocin remains critical in species with maternal-only care, such as sheep and rats[3,55]. Another but not mutually exclusive explanation is that mammalian parental care may originate from offspring-defense behaviors in anamniotes and thus shares the mechanism with the male-biased proceptive sexual behaviors (see ref. [23]). These issues will be addressed in future studies.

Within the same experiments as infant-retrieval assays, atosiban robustly inhibited physical contact of sibling marmosets with their family members. This finding aligns with oxytocin's

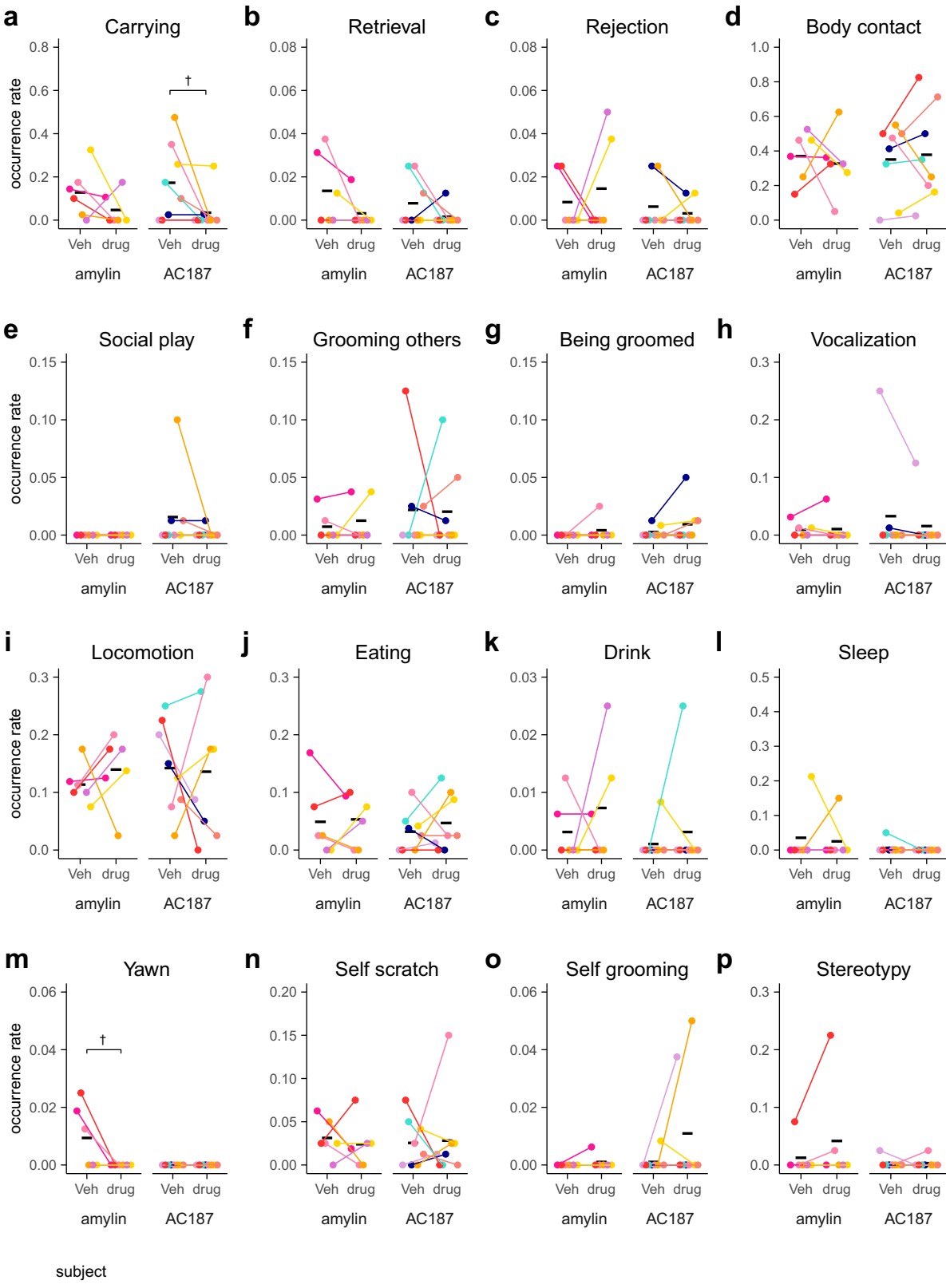

**Fig. 6 Effects of amylin and AC187 in the cMPOA on the family observation.** Various behaviors in family observation (**a–p**) after the vehicle (Veh), amylin or AC187 microinfusion. The family observation was conducted for 21 observations with amylin using 3 families and 20 observations with AC187 using 5 families (Umihei and Rika were observed simultaneously because they were littermates). The means of each subject in each condition are shown as dots and connected between two conditions. The black horizontal lines indicate the mean values of each condition of all subjects. Asterisks denotes statistical significance (paired $t$-test, amylin: control; $n = 11$ observations of 6 subjects, amylin; $n = 10$ observations of 6 subjects, AC187: control; $n = 12$ observations of 8 subjects, AC187; $n = 12$ observations of 8 subjects, $^*p < 0.05$, $^†p < 0.1$).

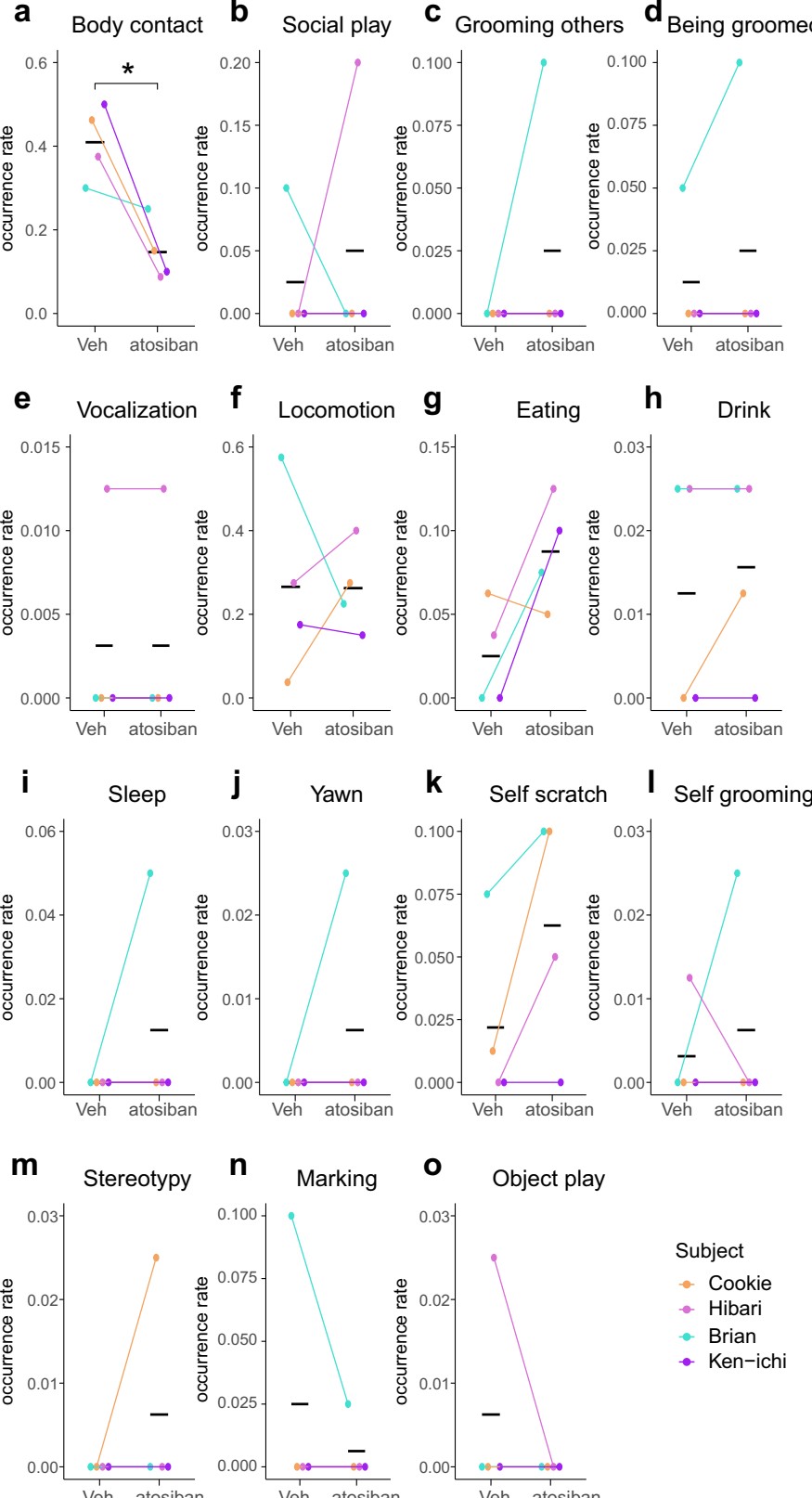

**Fig. 7 cMPOA infusion of atosiban, an inverse agonist of oxytocin reduces social contact with family members other than infants.** Various behaviors except for infant-directed behaviors observed in family observation (**a**–**o**) after the vehicle (Veh) or atosiban microinfusion. The means of each subject in each condition are shown as dots. The black horizontal lines indicate the mean values of each condition of all subjects. Asterisks denotes statistical significance (paired *t*-test, control; *n* = 6 observations of 4 subjects, atosiban; *n* = 6 observations of 4 subjects, *$p < 0.05$). Among behaviors listed in Table 1, marking and object play are omitted because of low occurrence. Refer to Supplementary Fig. 2 for infant-directed behaviors in the family observation and infant-retrieval assays on the same day.

involvement in social contact[37,38] and may also be relevant to its role in energy metabolism and cold defense[36,56,57]. Although the exact neural mechanisms for these effects need further investigation, these data at least imply that distinct neural and molecular mechanisms are involved in different types of social interactions, for which the cMPOA is a vital brain site.

Research on parental care will ultimately contribute to understanding the neurobiological factors causing parental stress and affecting parental motivation in humans, which may lead to child abuse and neglect. More research should be conducted on non-human primates to support parent-infant interactions and prevent child maltreatment. For this purpose, the common marmoset will serve as the valuable primate model of family parenting in humans with access to various neurobiological tools.

## Methods

**Animals**. All animal experimentation was approved by the Animal Experiment Judging Committee of RIKEN and was conducted in accordance with the National Research Council's "Guide for the Care and Use of Laboratory Animals, 8th Edition"[58]. Common marmosets were reared at the RIKEN Center for Brain Science in accordance with the institutional guidelines and under veterinarians' supervision.

All common marmosets we used were kept in a family at the RIKEN Wako campus, Japan. One marmoset breeding cage was 43 (width) × 66 (height) × 60 (depth) cm. Each cage contained a food tray, a water faucet, two wooden perches, and a metal mesh loft. Two or three cages were joined through a square hole (9.6 cm wide × 10.5 cm high) on a side panel or a metal mesh tunnel (75 cm wide × 30.5 cm tall × 21 cm deep) placed in front of two cages, depending on the number of family members in accordance with the ethical guideline of RIKEN, to form one single home cage. Although tactile contact was restricted between families, visual, olfactory, and auditory communication was possible in the colony room. Water and food pellets (CMS-1M, Clea Japan Inc., Japan) were supplied ad libitum. The monkeys' food was replenished at approximately 11:30. Supplementary foods, such as a piece of a sponge cake, dried fruits, and lactobacillus preparation, were replenished at approximately 16:00. The photoperiod of the colony room was 12 L:12 D (light period: 8:00–20:00, dark period: 20:00–8:00). The observations and experiments were conducted in the animals' home cages between 8:00 and 17:00. All marmosets were well habituated to the presence of the experimenters (TK, KS, and technical staff members) in the colony room to conduct the observations and experiments.

**Family observation assay**. Family observations were performed as described[23]. Briefly, all the family members were lured into the center and right compartments using a piece of palatable food. The left compartment was shuttered to restrict their location into the two cages for the ease of on-site behavioral coding (Fig. 1d). Then each family member's infant caretaking, other social and non-social behaviors listed in Table 2 were coded on-site for 20 min with 30-s bins with simultaneous recording using a video camera (HDR-AS100V, Sony, Tokyo, Japan). The behavioral coding by on-site observations was converted to "the occurrence rate" = (the number of bins in which each behavior was observed)/{total bins (40)}. The occurrence rates were used for analyses. It should be noted that the infant-carrying behaviors in the family observation are dependent on the behaviors of other family members and thus more variable than those in the dyadic infant-retrieval assays, which directly reflect the subject's voluntary activities toward a single infant[23].

**Infant retrieving assay**. The dyadic infant-retrieval assays (Fig. 1c) were performed as described[23]. Briefly, an infant, the subject's younger sibling (2–36 days old), was presented as the stimulus animal. Because there were generally twins in each family, the stimulus infants were counterbalanced between control (vehicle infusion) and experimental conditions. All subjects were acclimatized to the tunnel and the wire-mesh basket (15 cm diameter × 15 cm high) without infants before the test. By luring with palatable food, the subject was placed in the center cage, other family members were placed in the left cage, and the mesh shutter was set at the entrance of the center cage before the starting test. Then the stimulus infant was gently taken up from a carrier and placed into the mesh basket, which contained an electric hand warmer (KIR-SE1S, Sanyo, Osaka, Japan) covered with gauze to keep the infant warm during separation. The infant in the mesh basket was placed in the right cage. Opening the shutter of the caregiver's cage allowed the caregiver to access the infant's cage (Fig. 1c). The behavior of the caregivers and infants before retrieval and 600 s after retrieval, or for 600 s after the opening of the shutter when retrieval was not attempted, was directly observed and recorded using two video cameras (HDR-AS100V, Sony, Tokyo, Japan) as well as a directional microphone (MKH 416, Sennheiser, Hanover, Germany) connected to a linear PCM recorder (DR-60DMKII, Tascam, Tokyo, Japan). The audio was recorded at 24-bit and 96 kHz. The time from the opening of the sliding door to the retrieval of the infant, defined as when all of the infant's limbs were in contact with the caregiver's body, was recorded as the retrieval latency. Immediately after successful retrieval, the caregivers' infant-directed behaviors were coded on-site for 600 s with 30-s bins. On-site observation coded subjects' behavioral repertoire as listed in Table 2 except for body contact, social play, being groomed, and grooming others, because these behaviors will not occur in the absence of other family members in test cages. The results of these on-site observational records were used for analyses.

Two variables were calculated based on Shinozuka et al.[23]: (1) rejection rate, the number of bins that included rejection divided by the number of bins of infant carrying; (2) carrying rate, the duration of infant carrying (30× the number of bins of infant carrying) divided by the total length of the session (= retrieval latency + 600-s).

**MRI**. As a pretreatment for animals for magnetic resonance imaging (MRI), animals were intubated to maintain constant respiration under anesthesia with a mixture of oxygen and isoflurane (Abbott Laboratories, Abbott Park, IL, USA) using an artificial respirator (SN-480-7; Shinano, Tokyo, Japan). Animals were laid in a spine position during scanning, and the animal's physiological conditions were continuously monitored. MRI scans were obtained on a 9.4 T Biospec 94/30 MRI (Bruker BioSpin; Ettlingen, Germany) with a conventional linear polarized bird-cage resonator transmitter coil (Bruker BioSpin; inner diameter 86 mm) with actively shielded gradients at a maximum strength of 660 mT/m, as described[59]. T2-weighted images (T2WI) were acquired using a rapid acquisition with relaxation enhancement sequence with the following parameters: repetition time/echo time echo = 5000 ms/40.0 ms, rare factor = 4, field of view = 40 mm × 32 mm, matrix = 200 × 160, slice thickness = 0.2 mm, number of slices = 140, cross-section = coronal plane, number of averages = 8, scanning time = 33 min 20 s.

**Stereotaxic surgeries for cannula implantation and excitotoxic lesions**. Animals were deeply anesthetized with intramuscular injection of a mixture of xylazine (2.4 mg/kg) and ketamine (30 mg/kg), then with inhalation of isoflurane (1–3% in air).

The stereotaxic coordinates based on the interaural zero reference point were based on the marmoset brain atlas[33,60], and were adjusted using the pre-surgical MRI images whenever available.

Our preparatory experiments suggested that placing the bilateral 26-gauge guide cannulas may disturb the subjects' alloparental behaviors, especially when the guide cannula tips were close to (but still above) the anterior commissure. To minimize the tissue damage by the bilateral guide cannula, we next planned to implant the bilateral guide cannula much shallower (close to the corpus callosum). However, such short guide cannula may have a weaker guiding function so that the internal cannula could be off-target. To overcome these initial concerns, we prepared the following two different surgical groups. In hindsight, the shallow bilateral cannula worked well, while the unilateral cMPOA lesions partially disturbed the alloparental care.

Eight marmosets (3 males and 5 females, Table 1) received a stereotaxic surgery for the unilateral (right hemisphere) excitotoxic lesion and the contralateral (left) implantation of a guide cannula, made of either stainless or MRI-compatible polyetheretherketone (PEEK) resin (26 gauge; C311GS-5/SPC or C315GS-5/PK/SPC, Plastics One, Roanoke, VA, USA) (Fig. 1a, top). The 500 nl of 20 mg/ml saline excitotoxic amino acid, N-methyl-D-aspartic acid (NMDA; M3262, Sigma-Aldrich, St. Louis, MO, USA) was injected into the right cMPOA (A +10.8 mm, L −0.6 mm, V +7.3 mm) through a pulled glass capillary (tip diameter 20–50 μm) connected to a 5 μl Hamilton syringe with a microinjector (IMS-20, Narishige, Tokyo, Japan) by oil pressure under sterile conditions. Then a guide cannula was implanted into the left side, of which the tip was placed at A +10.8 mm, L +0.6 mm, V +11.3 mm (Kotaro, Hanako) or 12.0 mm (others). The internal cannula was designed to protrude 4 or 4.7 mm, respectively, from the tip of the guide cannula to target the cMPOA. The guide cannula was fixed by resin and dental cement, and a stainless or PEEK resin dummy cannula (C315DCS-5/SPC or C315DCNS-5/SPC; 10 or 10.8 mm below pedestal, Plastics One, Roanoke, VA, USA) was inserted into the guide cannula. The cannula was then covered by a truncated screw-capped tube for protection.

Nine marmosets (4 males and 5 females) received surgery for bilateral guide cannula (26 gauge; C235GS-5-1.2/SPC; 11.8 mm below pedestal, Plastics One) and dummy cannula (C235DCS-5-1.2/SPC, Plastics One) was inserted. The tip of the guide cannula was placed at A +10.8 mm, L ±0.6 mm, V +12.45 mm, and the protrusion of the internal cannula from the guide cannula was 5.15 mm. This modification was made because our preliminary experiments showed that parental behavior was disturbed by the deeper implantation of the bilateral guide cannula.

During the 2-day recovery period after surgery, marmosets that had undergone surgery were separated in one of the home cage complexes with visual/olfactory access to the family members through the mesh wall and treated with analgesics and antibiotics. Then they were reunited with the family before the maternal parturition and tested for various social and non-social behaviors. After the behavioral experiments, the microinfusion sites and the areas of fluid infusion were determined after trypan blue infusion (Supplementary Fig. 1).

The marmoset subjects analyzed in this study are listed in Table 1. Separate from them, several marmosets were excluded from the analysis after surgery due to the following reasons: mild bleeding from the guide cannula without general health problems (2); a post-surgical health problem (1); disturbed infant-care behaviors before experiments (1); an accidental injury unrelated to the experiment (1); the rejection and attack from the family (1); mistargeting of the cannula (1).

**Microinfusing pharmacological agents into the cMPOA.** Each marmoset was subjected to the microinfusion experiment 2–4 times per week, alternating the drug and its vehicle conditions during a single postpartum period of their mother (see Fig. 2d). After the birth of the infants in the family, the subject siblings were first verified for their normal infant-directed and intrafamilial behaviors without any infusion (intact). Then they were subjected to microinfusion experiments during the postnatal day 2–36. If the stimulus infants became mature for locomotion and unwilling to be carried, the last drug-vehicle condition pair were both excluded from the analyses.

Saline, artificial cerebrospinal fluid (aCSF; 3525/25ML, Tocris Bioscience, Bristol, UK), muscimol (M1523-5MG, Sigma-Aldrich, St. Louis, MO, USA; 0.2–0.6 μg/μl saline or aCSF), amylin (mouse, rat) (4030201, Bachem, Torrance, CA; 2 μg/μl aCSF), AC187 (amylin antagonist) (ab141150, Abcam, Cambridge, UK; 10–20 μg/μl aCSF), or atosiban (oxytocin inverse agonist) (A3480-10MG, Sigma-Aldrich, St. Louis, MO, USA; 12.5–50.0 ng/μl aCSF) was microinfused into the cMPOA. All drug infusions were paired with the infusion of its vehicle solution of the same amount, and the order of drug/vehicle infusions was counterbalanced among subjects. Each subject was injected with a single agent per day.

During the microinfusion, each subject was manually held, and the solution was infused through an injection cannula (35 gauge; custom-made, Saito Medical Instruments, Inc., Tokyo, Japan) connected to a 5 μl Hamilton syringe with a microinjector by oil pressure, targeting the cMPOA (A +10.8 mm, L ±0.6 mm, V +7.3 mm).

The first five marmosets with the unilateral cannula (Table 1) were examined by family observation and infant-retrieving assay only with muscimol microinfusion. 500 nl of muscimol or saline was injected at 11:00 over 10 min (50 nl/min). Two hours after infusion, the behavioral assay was started with family observation (Fig. 1b, middle).

The remaining three marmosets with unilateral cannulae (Nabe, Ringo and Senae) were used to seek appropriate experimental conditions; thus, the procedures were mixed, as detailed in Table 1.

For the nine marmosets with bilateral cannula, 100–500 nl of muscimol, the same amount of the vehicle, 500 nl of pharmacological agents or aCSF was injected between 11:00–13:00 over 1.5–7.5 min (66.6 nl/min). The behavioral assays were started 30 min after the infusion, first the infant-retrieving assay and then family observation, to accommodate the rapid degradation of amylin in the brain tissue[61].

**Brain tissue preparation for c-Fos expression mapping after infant care.** Two marmosets without surgical interventions were subjected to histological analyses after infant exposure. After one overnight isolation from the family, each subject was presented with an infant (the subject's younger sibling) and behavior was observed for 30 min. They carried the infant throughout without rejection. Then the infant was removed from each subject, and 75 min later, the subjects were anesthetized and perfused transcardially with 4% paraformaldehyde as described below.

**Preparation of brain sections.** Animals were deeply anesthetized first with intramuscular injection of a mixture of xylazine (2.4 mg/kg) and ketamine (30 mg/kg) and then with sodium pentobarbital (80 mg/kg, i.p.). In cannulated marmosets, 500 nl of trypan blue was injected through an injection cannula over 10 min to visualize the infusion site (Supplementary Fig. 1) before sodium pentobarbital injection. Then animals were perfused transcardially with 0.1 M phosphate buffer (PB, pH 7.4) then 4% (w/v) paraformaldehyde (PFA) in 0.1 M PB. The brains were

removed and immersed in the same fixative at 4 °C overnight, followed by cryoprotection in the series of 20%, 30% and another 30% (w/v) sucrose in PBS until they sank (typically 1, 2, 1–2 days for each sucrose solution, respectively), frozen by being buried in powder dry ice, and stored at –80 °C until sectioning. Brains were sectioned on a freezing microtome at 20 or 40 μm. Every 8th (Fig. 4a, b) or 6th (Fig. 4c, d) section from the serial sections was processed for immunohistochemistry (IHC). The brains of cannulated marmosets were sectioned at 40 μm, and every 2nd section was photographed to confirm the infusion site by trypan blue according to the marmoset brain atlas[33].

**Immunohistochemistry**. Immunohistochemistry (IHC) on free-floating sections was performed essentially as described[11]. The 20 μm brain sections were used for IHC. Antigen retrieval was performed with 0.01 M citrate buffer (pH 6) in 10% glycerol at 85 °C for 35 min then at room temperature for 30 min, after 1st wash with PBS containing 0.2% Triton-100 (PBST) for 3 h. The sections were washed with PBST, incubated with 0.3% $H_2O_2$ in methanol for 5 min, washed with PBST, blocked with 0.8% Block Ace (Dainihon-Seiyaku, Osaka, Japan), and 10% normal goat serum (for anti-Calcr and Amylin antibody) in PBST. The sections were incubated at 4 °C for 3 days with rabbit primary antibody against c-Fos (1:5000, sc-52, Santa Cruz Biotechnology, Inc., Dallas, TX, USA) (note: the staining results were notably varied by the antibody lot numbers) or against Amylin (1:10,000, H-017-11, Phoenix Pharmaceuticals, Burlingame, CA, USA). The following morning, the sections were washed and incubated with 50% EnVision+ Single Reagents, HRP. Rabbit in 0.4% Block Ace (K400311-2, Agilent Technologies, Inc. Santa Clara, CA, USA) for 1 h according to the manufacturer's instructions. The sections were immersed in 0.1 M boric buffer (pH 8.5) containing 4 μM Alexa488 or 568-labeled tyramide, 4% dextran sulfate, 0.05 mg/ml iodophenol, and 0.003% $H_2O_2$ for 30 min. Following the first staining, the sections were processed similarly second staining using rabbit anti-Calcr (1:250, AHP635, Bio-Rad Laboratories, Inc., Hercules, CA) or goat anti-NeurophysinI (NPI (1:5000, sc-7810, Santa Cruz Biotechnology). The pink-red color was developed by 30 min of immersion in ImmPACT Vector Red substrate (ImmPACT Vector Red AP Substrate Kit, Vector Laboratories). Subsequently, they were washed with PBS and then mounted on gelatin-coated slides using a mounting medium (Vectashield; Vector Laboratories). Photomicrographs were taken using an inverted microscope (KEYENCE BZ-X700, Osaka, Japan) with a 10–40× objective. The contrast and brightness of all photographs were adjusted only linearly and uniformly for all the micrographs used in one experiment, using software (Image J[62] or BZ-X Analyzer (KEYENCE BZ-X700, Osaka, Japan)).

**In situ hybridization**. In situ hybridization was performed using the RNAscope® platform according to the manufacturer's instructions (Advanced Cell Diagnostics (ACD), Newark, CA, USA). The marmoset Amylin mRNA-specific probe (1154171-C1, RNAscope™ Probe- Cj-IAPP-O1-C, Advanced Cell Diagnostics (ACD), Newark, CA, USA) was used together with a single detection kit (RNAscope® 2.5 HD Reagent Kit-RED, ACD, Cat# 322350). In brief, the 20-μm-thick brain sections were mounted on MAS-coated slides and dried for 1 h at room temperature. The sections were pretreated with hydrogen peroxide included in the kit for 10 min, 1X Target Retrieval Buffer (ACD) for 15 min and protease plus (ACD) for 30 min and incubated with a probe solution for 2 h at 40 °C in the HybEZTM oven (ACD). The signal was amplified following the manufacturer's recommendation (AMP1~6, BROWN-A and -B), followed by counterstaining with Hematoxylin Solution, Gill No. 3 (GHS316,

Merck, Darmstadt, Germany) and mounted with Softmount (199-11311, FUJIFILM Wako Pure Chemical Corporation). Photomicrographs were taken using an inverted microscope (KEYENCE BZ-X700, Osaka, Japan) with a 10–40× objective. The contrast and brightness of all photographs were adjusted only linearly and uniformly for all the micrographs used in one experiment, using software (Image J[62] or BZ-X Analyzer (KEYENCE BZ-X700, Osaka, Japan)).

**Statistics and reproducibility**. We used software R[63] for statistical analyses. All behaviors observed with muscimol microinfusion (Figs. 2 and 3) were analyzed using the generalized linear mixed model (GLMM). In these models, the microinfusion type (muscimol or vehicle), the cannula type (unilateral or bilateral) and their interactions were used as explanatory variables, and the subject as a random effect to control pseudoreplication. We used GLMM with normal distribution, including those behaviors with very low occurrence rates (e.g., yawning) and thus showed overdispersion. The reasons for this are, firstly, GLMM with binominal distribution (which should be used for behavioral counts within a fixed number of bins) could not yield an appropriate fit due to its inherent vulnerability to overdispersion (as in the binominal distribution, the dispersion must be equal with sample size × occurrence rate × (1 – occurrence rate)), and second, it is preferable to consistently use the statistical method for all these naturalistic behaviors. The "lme4" package was used for GLMM, the "car" package for evaluation of the significance of the explanatory variables, and the "emmeans" package with Holm adjustment for multiple comparisons.

In the analyses of the effects of amylin, AC187 and atosiban, the GLMM model could not be reliably applied because of singular fit errors. Therefore, a paired $t$-test using mean values of each subject in each condition was applied for the analyses of infant retrieving and family observation assays instead of GLMM.

**Reporting summary**. Further information on research design is available in the Nature Portfolio Reporting Summary linked to this article.

## Data availability

All data needed to evaluate the conclusions in the paper are present in the paper and Supplementary Information. The source data behind the graphs can be found in Supplementary Data. The raw video and data files can be provided by K.O.K. (kurodalab@bio.titech.ac.jp) pending a material transfer agreement due to ethical restrictions.

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

## Acknowledgements

We thank Dr Michael Numan for the fruitful discussion, Sayaka Shindo and Yumi Ogawa for technical support, and the RIKEN Center for Brain Science, Research Resources Division for animal husbandry and care. This research was supported by RIKEN Center for Brain Science (2014-2023) to K.O.K., the Japan Agency of Medical Research and Development (AMED) under grant numbers JP20dm0107144 to K.O.K., JP22dm0207001, the program for Brain Mapping by Integrated Neurotechnologies for Disease Studies (Brain/MINDS) to H.O., and No. 16 dm0207003h0003 to K.N.; JSPS KAKENHI grant number JP20K16633 to T.K., JP20K12587 to K.S., JP19K16901 to S.Y.-N., 22K12236 to C.Y., and JP18KT0036 and JP22K19486 to K.O.K.; Takeda Science Foundation 2023028525 to K.O.K.

## Author contributions

T.K. and K.S. designed, carried out experiments, and produced the tables and figures with support from S.Y.-N., A.Y.M, and K.O.K. C.Y. carried out the IHC and ISH described in Fig. 4. J.H. and Y.H. carried out MRI scans of subjects under the supervision of H.O. K.O.K. conceived of and organized the study with T.K., K.S., and S.Y.-N., and wrote the manuscript with T.K., with contributions from all the authors.

## Competing interests

The authors declare no competing interests.

## Additional information



