## [Peer Review File · Communications Biology]

REVIEWERS' COMMENTS:

Reviewer #1 (Remarks to the Author):

The authors have addressed most of my concerns and I believe this paper is suitable for publication. However, I strongly recommend the authors add further clarifying language on the logic of their neuronal ablation control experiment. It seems like this is an important control experiment that allows the anterior commissure to remain intact (minimizing effects on behavioral baseline), but that logic is not explained well or immediately obvious to someone who does not commonly do primate surgeries. It is also unclear why this is a better control than a sham surgery. If I am incorrect in my understanding, I think this only further highlights the need for a better explanation.

Reviewer #2 (Remarks to the Author):

The present study focuses on the evolutionary conserved and specific role of Amylin and its receptor Calcitonin in infant caregiving behavior. The authors show that calcitonin receptor expressing neurons in the central medial preoptic area (cMPOA) are important for mediating infant caregiving in a primate model, common marmosets. Instead of the canonical lesion behavior studies in marmosets, the authors have used reversible pharmacological manipulations to inactivate neurons in the cMPOA. This manipulation reduced infant caring in the subadult marmosets sparing their social behavior with other family members and other non-social behaviors. Infusion of amylin in the MPOA promoted infant caring whereas oxytocin's inverse agonist, atosiban, affected interactions with the other family members. Altogether, the study hints at the crucial role of amylin-calcitonin and oxytocin in regulating social interactions in a family setting in marmosets.

Overall, the authors have been very responsive to critiques and the resulting revised manuscript is substantially strengthened. I have only one minor concern, that a new sentence added to the discussion is slightly confusing and should be revised for clarity:

"Another but not mutually exclusive explanation is that mammalian parental care stems from sex-unbiased (or even male-biased) reproductive behaviors in amniotes and thus oxytocin-independent originally; then along with the evolution of amniotic membranes, maternal-biased offspring emerged and became regulated by oxytocin for its synergistic activation with oviposition 53-55. "

Reviewer #3 (Remarks to the Author):

I'm not qualified to evaluate most of the methods used in the study, so, as in my previous review, I limit my comments to the behavioral methods/results and the writing in general.

I still think the first 3 paragraphs of the results should be moved to the methods section, but I understand that the authors prefer not to do this, although I'm not sure why.

Table 2 – As I mentioned in my previous review, more information should be given in Table 2, including the subjects' ages, which family each subject was in, how large the family was, and how much alloparental experience each subject had.

In the new paragraph addressing potential sex differences (L171-173), please clarify whether the sexes were compared statistically. I assume not, due to the small sample sizes, but this should be stated explicitly. Also, I don't find the color coding in Figs. 2 and 3 to be at all helpful with respect to

potential sex differences, since the colors don't clearly indicate which data are from which sex.

L 44 – Change “to” to “in”.

L72 – Change to either “increased postpartum” or “increased in the postpartum period”.

L78 – “Then” doesn't fit well here because it doesn't directly follow the mention of the previous study.

L80 – Early studies found that marmoset groups sometimes contain unrelated adults. This is worth mentioning.

L82 – Change to either “share the infant carrying” or “share the infant-carrying care”.

L90-93 – Reference for this sentence?

L98 – Briefly state what these limitations are.

The following phrases should be hyphenated (because they are compound adjectives preceding the noun they describe):

L82 – infant-carrying

L135 – infant-care

L177 – infant-retrieval

L186 – infant-care

Section on “Histological analysis of the marmoset MPOA” (p. 7) – The text comparing marmosets to mice would be more appropriate as part of the discussion.

L187 – Not clear what “their” refers to.

L223-224 – What was the direction of the difference in rejection rate (i.e., in which condition were rates higher)?

L252 – Reference for the statement about neurotoxic lesions?

L257 – Do you mean to say that lesions of this pathway (rather than the pathway itself) facilitate pup retrieval?

L263-264 – Reference?

L273 – It is premature to say that these techniques “should... conclusively demonstrate the selective role of...” The authors may hope and expect that these techniques will conclusively demonstrate this role, but it's not a foregone conclusion that this will happen.

L276-277 – It would be more appropriate to refer to “female group living” (or “female sociality”), since “group housing” generally refers to how they are housed in captivity, not how they choose to live in the field.

L281-L282 – I don't understand why this is "presumably related to self-defense type of social behaviors." The evolutionary drivers of marmoset sociality might involve access to mating opportunities, access to food resources, predator avoidance, or perhaps other factors, not just "self-defense social behaviors."

Kurachi et al., “Distinct roles of amylin and oxytocin signaling in within-family social behaviors at the medial preoptic area of common marmosets”

Referees' comments and our point-by-point responses

*Please note that, in the revised main text, we used color-coding: **Red** colored changes were for Reviewer #1's comments, **blue** for Reviewer #2's, **green** for Reviewer #3's comments, and **brown** for other changes, such as those by proofreading. For the changes attending two reviewers' comments, both colors were used in a half-and-half manner.

Reviewer #1

The authors have addressed most of my concerns and I believe this paper is suitable for publication. However, I strongly recommend the authors add further clarifying language on the logic of their neuronal ablation control experiment. It seems like this is an important control experiment that allows the anterior commissure to remain intact (minimizing effects on behavioral baseline), but that logic is not explained well or immediately obvious to someone who does not commonly do primate surgeries. It is also unclear why this is a better control than a sham surgery. If I am incorrect in my understanding, I think this only further highlights the need for a better explanation.

Response: Sorry for being still unclear. We have added the explanation in the “Stereotaxic surgeries for cannula implantation” in the Materials and Methods as follows; “**Our preparatory experiments suggested that placing the bilateral 26-gauge guide cannulas may disturb the subjects' alloparental behaviors, especially when the guide cannula tips were close to (but still above) the anterior commissure. To minimize the tissue damage by the bilateral guide cannula, we next planned to implant the bilateral guide cannula much shallower (close to the corpus callosum). However such short guide cannula may have a weaker guiding function, so that the internal cannula could be off-target. To overcome these initial concerns, we prepared the following two different surgical groups. In hindsight, the shallow bilateral cannula worked well, while the unilateral cMPOA lesions partially disturbed the alloparental care.**”

Thank you very much again for your careful review and constructive comments of our manuscript.

Reviewer #2 (Remarks to the Author):

The present study focuses on the evolutionary conserved and specific role of Amylin and its receptor Calcitonin in infant caregiving behavior. The authors show that calcitonin receptor expressing

neurons in the central medial preoptic area (cMPOA) are important for mediating infant caregiving in a primate model, common marmosets. Instead of the canonical lesion behavior studies in marmosets, the authors have used reversible pharmacological manipulations to inactivate neurons in the cMPOA. This manipulation reduced infant caring in the subadult marmosets sparing their social behavior with other family members and other non-social behaviors. Infusion of amylin in the MPOA promoted infant caring whereas oxytocin's inverse agonist, atosiban, affected interactions with the other family members. Altogether, the study hints at the crucial role of amylin-calcitonin and oxytocin in regulating social interactions in a family setting in marmosets.

Overall, the authors have been very responsive to critiques and the resulting revised manuscript is substantially strengthened. I have only one minor concern, that a new sentence added to the discussion is slightly confusing and should be revised for clarity: "Another but not mutually exclusive explanation is that mammalian parental care stems from sex-unbiased (or even male-biased) reproductive behaviors in amniotes and thus oxytocin-independent originally; then along with the evolution of amniotic membranes, maternal-biased offspring emerged and became regulated by oxytocin for its synergistic activation with oviposition 53-55. "

Response: We agree that this sentence was not clear enough, and revised it as “Another but not mutually exclusive explanation is that mammalian parental care may originate from sex-unbiased reproductive behaviors in amniotes⁵⁶. These issues should be addressed in future studies.”.

Thank you very much again for your careful review and constructive comments of our manuscript.

Reviewer #3 (Remarks to the Author):

I'm not qualified to evaluate most of the methods used in the study, so, as in my previous review, I limit my comments to the behavioral methods/results and the writing in general.

I still think the first 3 paragraphs of the results should be moved to the methods section, but I understand that the authors prefer not to do this, although I'm not sure why.

Response: Sorry for being unclear. This is simply because of the order of sections in this journal.

Traditionally, a scientific manuscript consists of

Introduction -> Methods -> Results -> Discussion.

However, this journal has the style:

Introduction -> Results -> Discussion -> Methods.

In this style, it is unavoidable to explain some experimental methods during the Results, because the readers seldom go back to read the Methods before going to the Results section. Still, we moved some

detailed explanation about the surgical procedures to the Methods section (colored red in the Method section).

Table 2 – As I mentioned in my previous review, more information should be given in Table 2, including the subjects' ages, which family each subject was in, how large the family was, and how much alloparental experience each subject had.

Response: We added subjects' age at the start of the behavioral experiments, previous experiences as caregivers, and family members present in the family observation cage in the revised Table 1 (formerly Table 1). The size of the family cage is described in the Methods.

In the new paragraph addressing potential sex differences (L171-173), please clarify whether the sexes were compared statistically. I assume not, due to the small sample sizes, but this should be stated explicitly. Also, I don't find the color coding in Figs. 2 and 3 to be at all helpful with respect to potential sex differences, since the colors don't clearly indicate which data are from which sex.

Response: We revised the mentioned paragraph that “Of additional note, we examined the sex difference of the effects of muscimol using the models including sex as an explanatory parameter together with drug and cannulat type and their interactions. We did not observe apparent sex differences and interactions in most behaviors except those with very low occurrence frequency (Supplementary Table 4 and 5), in harmony with our previous study²³. Therefore, we adopted the models excluding the sex effect.”

L 44 – Change “to” to “in”.

Response: We modified it as suggested.

L72 – Change to either “increased postpartum” or “increased in the postpartum period”.

Response: We added “period” in this sentence.

L78 – “Then” doesn't fit well here because it doesn't directly follow the mention of the previous study.

Response: We delete “then” from this sentence.

L80 – Early studies found that marmoset groups sometimes contain unrelated adults. This is worth mentioning.

Response: We have added the following phrase in the mentioned sentence, as “Common marmosets

live in a family that consists of a breeding pair (mother and father) and their offspring, **occasionally with unrelated adults in the wild** ²¹.”

L82 – Change to either “share the infant carrying” or “share the infant-carrying care”.

Response: We deleted “care” from this sentence.

L90-93 – Reference for this sentence?

Response: We have added the reference No.23.

L98 – Briefly state what these limitations are.

Response: We have added that “(e.g., **permanent brain dysfunctions may induce functional adaptation and remodeling of the neighboring brain areas** ^{26 27}). They thus should be complemented with data from reversible suppression of the function.”.

The following phrases should be hyphenated (because they are compound adjectives preceding the noun they describe):

L82 – infant-carrying

L135 – infant-care

L177 – infant-retrieval

L186 – infant-care

Response: All of those words were modified as mentioned above.

Section on “Histological analysis of the marmoset MPOA” (p. 7) – The text comparing marmosets to mice would be more appropriate as part of the discussion.

Response: We have moved “**in mice, the anterior extension of the aPVH is termed the anterior commissural nucleus (AC). The AC possesses distinctive features from the aPVH in cell morphology and its transcriptional activation in non-oxytocinergic neurons by performing parental care. Thus, this part of the aPVH in marmosets may functionally correspond to the AC in mice.**” From the mentioned part to the Discussion.

L187 – Not clear what “their” refers to.

Response: We replaced “their” to “**the amylin’s**”.

L223-224 – What was the direction of the difference in rejection rate (i.e., in which condition were rates higher)?

Response: Thank you for pointing this issue. We replaced “the difference between the rejection rate” to “**the increase of** the rejection rate”.

L252 – Reference for the statement about neurotoxic lesions?

Response: We have added the reference No.23.

L257 – Do you mean to say that lesions of this pathway (rather than the pathway itself) facilitate pup retrieval?

Response: No. Lecca et al shows that activation of BST-innervation to LHb neurons promotes pup retrieval, and inhibition of them inhibit pup retrieval (Fig. 4A-D, Lecca et al, 2023).

L263-264 – Reference?

Response: We have added two references (No.13, 18) here.

L273 – It is premature to say that these techniques “should... conclusively demonstrate the selective role of...” The authors may hope and expect that these techniques will conclusively demonstrate this role, but it’s not a foregone conclusion that this will happen.

Response: We have amended this sentence as “In the future, the cell-type or molecular-specific manipulation technique should become available in marmosets as in mice and **may better elucidate** the role of amylin-CalcR signaling in the MPOA.”.

L276-277 – It would be more appropriate to refer to “female group living” (or “female sociality”), since “group housing” generally refers to how they are housed in captivity, not how they choose to live in the field.

Response: We modified it as “female group living”, and we also modified “group housing” in L74.

L281-L282 – I don’t understand why this is “presumably related to self-defense type of social behaviors.” The evolutionary drivers of marmoset sociality might involve access to mating

opportunities, access to food resources, predator avoidance, or perhaps other factors, not just “self-defense social behaviors.”

Response: Sorry for being unclear. We replaced the sentence to directly cite the original proposal made by Eibl-Eibesfeldt, as “Eibl-Eibesfeldt suggested that mammalian sociality has two principal motives: “parental-drive” and “flight-drive”⁴⁶. We have demonstrated that while parental motivation activates and depends on the cMPOA, defensive huddling against a sudden bright light (an example of “flight”-driven social contacts) does not in mice¹⁸. These findings suggest that the within-family social behaviors of sibling marmosets (targeted primarily toward their parents) are originated from a distinct motivation from infant care. The neural basis of such within-family social motivation remains to be identified.”

Other changes: (Major changes were colored in orange-brown)

We realized that data from one animal (Ken-ichi) was not included in Fig. 3. We thus redid the figure and the analysis, and marked the changes by this color in the Results section. There is no significant alterations in the results of main effects.

We used software image J to analyze photographs of IHC and ISH, but we missed to refer it in the previous version. In this version, we added it in reference list.

We noticed that, in our subjects, some marmosets were too old to be called “subadult”. We modified the words as “sibling” in main text.

We have amended one description in the “Statistical analyses” of the Materials and Methods section to be more precise, from “as in the binominal distribution, the average must be equal with the dispersion”, to “as in the binominal distribution, the dispersion must be equal with sample size \times occurrence rate \times (1 – occurrence rate)”.

We modified Abstract section as “Calcitonin receptor (Calcr) and its brain ligand amylin in the medial preoptic area (MPOA) are found to be critically involved in infant care and social contact behaviors in mice. In primates, however, the evidence is limited to a lesion study of the Calcr-expressing MPOA subregion (cMPOA) in a family-living primate species, the common marmoset. The present study

utilized pharmacological manipulations of the cMPOA and shows that reversible inactivation of the cMPOA abolishes infant-care behaviors in sibling marmosets without affecting other social or non-social behaviors. Amylin-expressing neurons in the marmoset MPOA are distributed in the vicinity of oxytocin neurons in the anterior paraventricular nucleus of the hypothalamus. While amylin infusion facilitates infant-carrying selectively, an oxytocin's inverse agonist, atosiban, reduces physical contact with non-infant family members without grossly affecting infant care. These data suggest that the amylin and oxytocin signaling mediate intrafamilial social interactions in a complementary manner in marmosets.” because our abstract were more than 150 words.

We modified the acronym “BST” as “the bed nucleus of stria terminalis” because this acronym was used in main text less than three times.

We added Table S6 in L288 and Table S7 in L232 because we had forgot to cite these Tables.

We modified the section “Statistical analysis” as “Statistics and Reproducibility” according to formatting guideline.

We modified the section “Conflict of interests” as “Competing interests” according to formatting guideline.

We modified all supplementary figures and tables as “Supplementary Fig. 1”/”Supplementary Table 1” from “Fig. S1”/”Table S1” according to formatting guideline.

We modified figure caption in Fig. 4 B) and D) “cPVN” as “aPVH”.

We performed grammatical/spelling check again.